# Comparability between Computed Tomography Morphological Vascular Parameters and Echocardiography for the Assessment of Pulmonary Hypertension in Patients with Severe Aortic Valve Stenosis—Results of a Multi-Center Study

**DOI:** 10.3390/diagnostics12102363

**Published:** 2022-09-29

**Authors:** Elke Boxhammer, Bernhard Scharinger, Reinhard Kaufmann, Herwig Brandtner, Lukas Schmidbauer, Jürgen Kammler, Jörg Kellermair, Christian Reiter, Kaveh Akbari, Matthias Hammerer, Hermann Blessberger, Clemens Steinwender, Klaus Hergan, Uta C. Hoppe, Michael Lichtenauer, Stefan Hecht

**Affiliations:** 1Department of Internal Medicine II, Division of Cardiology, Paracelsus Medical University of Salzburg, Müllner Hauptstraße 48, 5020 Salzburg, Austria; 2Department of Radiology, Paracelsus Medical University of Salzburg, 5020 Salzburg, Austria; 3Department of Cardiology, Kepler University Hospital, Medical Faculty of the Johannes Kepler University Linz, 4020 Linz, Austria; 4Department of Radiology, Johannes Kepler University Hospital Linz, 4020 Linz, Austria

**Keywords:** aortic valve stenosis, computed tomography, pulmonary hypertension, systolic pulmonary artery pressure, TAVR

## Abstract

Background: Computed tomography (CT) of the aorta and cardiac vessels, which is performed in patients with severe aortic valve stenosis (AS) before transcatheter aortic valve replacement (TAVR), offers the possibility of non-invasive detection of pulmonary hypertension (PH), for example, by determining the diameter of the main pulmonary artery (PA), the right pulmonary artery (RPA) or the left pulmonary artery (LPA). An improvement of the significance of these radiological parameters is often achieved by indexing to the body surface area (BSA). The aim of this study was to compare different echocardiographic systolic pulmonary artery pressure (sPAP) values with radiological data in order to define potential clinical cut-off values for the presence or absence of PH. Methods: A total of 138 patients with severe AS undergoing TAVR underwent pre-interventional transthoracic echocardiography with determination of sPAP values and performance of CT angiography (CTA) of the aorta and femoral arteries. Radiologically, the PA, RPA, LPA, and ascending aorta (AA) diameters were obtained. Vascular diameters were not only indexed to BSA but also ratios were created with AA diameter (for example PA/AA-ratio). From these CT-derived vascular parameters, AUROC curves were obtained regarding the prediction of different sPAP values (sPAP 40–45–50 mmHg) and finally correlation analyses were calculated. Results: The best AUROC and correlation analyses were generally obtained at an sPAP ≥ 40 mmHg. When considering diameters alone, the PA diameter was superior to the RPA and LPA. Indexing to BSA generally increased the diagnostic quality of the parameters, and finally, in a synopsis of all results, PA/BSA had the best AUC 0.741 (95% CI 0.646–0. 836; *p* < 0.001; YI 0.39; sensitivity 0.87; specificity 0.52) and Spearman’s correlation coefficient (r = 0.408; *p* < 0.001) at an sPAP of ≥40 mmHg. Conclusions: Features related to pulmonary hypertension are fast and easily measurable on pre-TAVR CT and offer great potential regarding non-invasive detection of pulmonary hypertension in patients with severe AS and can support the echocardiographic diagnosis. In this study, the diameter of the main pulmonary artery with the additionally determined ratios were superior to the values of the right and left pulmonary artery. Additional indexing to body surface area and thus further individualization of the parameters with respect to height and weight can further improve the diagnostic quality.

## 1. Introduction

Pre-interventional computed tomography (CT) diagnostics in patients with severe aortic valve stenosis (AS) is currently standard practice for planning a transcatheter aortic valve replacement (TAVR) procedure [1]. In this context, not only the evaluation of the vascular structures in the groin (minimal vessel diameter, tortuosity, calcification of the iliac and femoral arteries) plays an important role for optimal access planning, but also the determination of the diameter of the aortic valve annulus to determine the size of the prosthesis to be implanted [2].

In addition to accurate preoperative/pre-interventional planning, CT diagnostics are also able provide evidence for the presence of pulmonary hypertension (PH) in patients with severe AS [3,4,5]. In the 2015 European Society for Cardiology (ESC) guidelines [6], a diameter of the main pulmonary artery (PA) ≥ 29 mm is considered potentially indicative of the presence of pulmonary hypertension, which is associated with a poor prognosis in terms of increased mortality in patients with severe AS. The most recent ESC guidelines from august 2022 [7] suggest the pulmonary artery to ascending aorta ratio (PA/AA-ratio) of 0.90 for the presence of PH, lowering the PA/AA-ratio by 0.10 compared to the 2015 guidelines. In addition, other vascular parameters or ratios are mentioned in the literature that are not specifically mentioned in the ESC guidelines.

Since right heart catheterization is no longer routinely performed prior to TAVR, transthoracic echocardiography is currently used in clinical practice as the gold standard to assess the presence of PH [8,9]. In most cases, the systolic pulmonary artery pressure (sPAP) is determined, which consists of the determination of the maximal continuous wave Doppler regurgitation velocity across the tricuspid valve and the estimation of the central venous pressure via the diameter and respiratory variability of the inferior vena cava. However, echocardiography is a procedure that is highly dependent on the experience of the examiner on the one hand and on the sound quality of the patient on the other.

Thus, the aim of the retrospective multi-center study design presented here was to compare CT morphological vascular parameters potentially associated with PH with different echocardiographically obtained sPAP values in order to draw further noninvasive conclusions about PH in TAVR patients. On the one hand, this is to strengthen the validity of echocardiography and, on the other hand, to establish cut-off values for various radiological vascular parameters in TAVR patients with PH.

## 2. Material and Methods

### 2.1. Study Population

This study population originally included 163 patients undergoing TAVR procedure between 2016 and 2018 at Paracelsus Medical University Hospital Salzburg and Kepler University Hospital Linz. Twenty-five patients had to be excluded due to missing weight or height data, missing CT data, or inadequate CT quality. Finally, 138 patients were recommended for inclusion in the study. The study protocol was authorized by the local ethics committees of Paracelsus Medical University Salzburg (415-E/1969/5-2016) and Johannes Kepler University Linz (E-41-16) and conducted in accordance with principles of the Declaration of Helsinki and Good Clinical Practice. Written informed consent was available from all study participants.

### 2.2. Transthoracic Echocardiography

Transthoracic echocardiography was routinely performed before TAVR by using common ultrasound devices (iE33 and Epiq 5; Philips Healthcare, Hamburg, Germany).

Severe AS was categorized according to current guidelines of ESC using an AV Vmax (maximal velocity over aortic valve) of 4.0 m/s, an AV dpmean (mean pressure gradient over aortic valve) ≥ 40 mmHg and an aortic valve area ≤ 1.0 cm^2^ for definition of severe AS. Left ventricular ejection fraction (LVEF) was calculated by usage of the Simpson’s method. Graduation of mitral, aortic, and tricuspid valve regurgitation in minimal, mild (I), moderate (II), and severe (III) was done by spectral and color-Doppler images. Maximum tricuspid regurgitant jet velocity (TRV) was obtained by continuous wave Doppler over the tricuspid valve. Pulmonary artery pressure (PAP) was calculated using the formula 4 × TRV^2^. To this value the estimated right atrial pressure (RAP) was added, which was determined by the diameter of the inferior vena cava (IVC). An IVC diameter ≥ 21 mm and a respiratory caliber fluctuation < 50% led to a RAP of 15 mmHg. For an IVC diameter < 21 mm as well as a respiratory caliber fluctuation ≥ 50%, a RAP of 3 mmHg was assumed. Other scenarios not corresponding to these constellations were provided with an intermediate value of 8 mmHg. The simplified Bernoulli equation (4 × TRV^2^) + RAP was applied to obtain a sPAP result. Different sPAP (40, 45 and 50 mmHg) values were used to determine PH in accordance with the current literature [10,11,12,13,14]. An illustration about the echocardiographic measurement is provided in Figure 1.

### 2.3. CTA Protocol and Measurement of Diameters for PH Assessment

Study patients at both centers—the University Hospital Salzburg and the Kepler University Linz—routinely received a pre-interventional, ECG triggered CT angiography (CTA) of the whole aorta and femoral arteries to assess, among others, the aortic annulus size, the aortic anatomy and vascular access. CT scans were performed on multidetector CT scanners (Salzburg: Somatom Definition AS+, Siemens Healthcare, Erlangen, Germany; Linz: Brilliance 64, Philips Healthcare, Hamburg, Germany) with a patient size-adapted tube voltage (80–120 kVp) and active tube current modulation. A bolus-tracking technique was applied with a 100 mL bolus of non-ionic iodinated contrast media followed by 70 mL saline solution injected at a flow rate of 3.5–5 mL/s.

A stationary workstation (Impax, Agfa-Gevaert, Mortsel, Belgium) was used for image analysis. Two experienced investigators, blinded to all clinical and hemodynamic information, performed the following measurements in mediastinal window settings on axial vessel cross sections on double oblique CT angiographic images as recommended previously [15,16]:The widest short-axis diameter of the main pulmonary artery (PA) within 3 cm of the bifurcation of the pulmonary trunk.The widest short-axis diameter of the ascending aorta (AA) at the level of the bifurcation of the pulmonary trunk.The widest short-axis diameter of the right pulmonary artery (RPA).The widest short-axis diameter of the left pulmonary artery (LPA).

An illustration of the measurements performed and the corresponding placements is shown in Figure 2. Using these measurements, the quotients of PA/AA-ratio, RPA/AA-ratio, and LPA/AA-ratio were formed. Furthermore, the diameters of PA, RPA, LPA, PA/AA, RPA/AA, and LPA/AA were index on body surface area (BSA) using the Du Bois formula (BSA = 0.007184 × Height^0.725^ × Weight^0.425^), thus providing PA/BSA, RPA/BSA, LPA/BSA, PA/AA/BSA, RPA/AA/BSA, and LPA/AA/BSA.

### 2.4. Statistical Analysis

Statistical analysis with graphical representations was performed using SPSS (Version 25.0, SPSSS Inc., Armonk, NY, USA).

A Kolmogorov–Smirnov test was carried out to test variables for normal distribution. Normally distributed metric data was expressed as mean ± standard deviation (SD). Not-normally distributed metric data was expressed as median and interquartile range (IQR). Frequencies/percentages were used for categorial data and compared using the chi-square test.

Area Under the Receiver Operator Characteristics (AUROC) curves with Area Under the Curve (AUC) were performed using different sPAP values (sPAP 40–45–50 mmHg) to obtain the cut-off values for the respective, CT morphological vascular parameters. In addition, sensitivity, specificity, and Youden Index (YI) were calculated separately.

Correlation analyses were performed using Spearman’s rank-correlation coefficient to determine the strength between different sPAP-values (again sPAP 40–45–50 mmHg) to different CT-measurements mentioned above (PA, RPA, LPA and ratios etc.).

A *p*-value < 0.050 was considered statistically significant.

## 3. Results

### 3.1. Baseline Characteristics

Table 1 gives an overview of the collected baseline characteristics of the included patient collective, whereas Table 2 once again tabulates the CT morphological vascular parameters obtained in relation to the respective sPAP values (sPAP ≥ 40–45–50 mmHg).

Regarding the demographics, 52.2% of the total cohort were male, with an overall mean age of 82.98 ± 5.05 years; 22.5% of the patients had diabetes mellitus, 81.9% had arterial hypertension, 34.1% had atrial fibrillation, and 8.0% had COPD. Echocardiography documented an average sPAP of 44.94 ± 16.65 mmHg. In this regard, 63.0% of patients had an sPAP of ≥40 mmHg and 50.0% had an sPAP ≥ 45 mmHg. CT morphology measured the main pulmonary artery (PA) to be 29.89 ± 5.45 mm and the ascending aorta (AA) to be 35.10 ± 4.61 mm. A PA diameter ≥ 29 mm according to the 2015 ESC guidelines [6] showed 60.9%, a PA/AA ratio ≥ 0.80, 62.6% and only 35.7% a PA/AA ratio ≥ 0.90, whereby the cut-off value ≥ 0.90 would correspond to the most current guidelines from 2022 [7].

### 3.2. AUROC Results—sPAP and PA (±Ratios)

In Figure 3, results of AUROC analysis with the aim to determine corresponding cut-off values for PA, PA/BSA, PA/AA, and PA/AA/BSA based on different sPAP values (sPAP 40–45–50 mmHg) are represented.

The best results in the present constellation were obtained at an sPAP of ≥40 mmHg (Figure 3A), first, for the ratio of PA/BSA, with a cut off value of 14.84 mm/m^2^ (AUC 0.741; 95% CI 0.646–0. 836; *p* < 0.001; YI 0.39; sensitivity 0.87; specificity 0.52) and secondly for the ratio from PA/AA/BSA with a cut off value of 0.42 (AUC 0.735; 95% CI 0.633–0.837; *p* < 0.001; YI 0.46; sensitivity 0.87; specificity 0.59). Another AUC value ≥ 0.700 was provided by the PA/AA ratio with a cut off value of 0.80 (AUC 0.704; 95% CI 0.603–0.804; *p* < 0.001; YI 0.36; sensitivity 0.75; specificity 0.61).

At an sPAP of ≥45 (Figure 3B) or ≥50 mmHg (Figure 3C), significant AUROC results were seen almost throughout, but these no longer reached the diagnostic quality as with an sPAP ≥ 40 mmHg.

### 3.3. AUROC Results—sPAP and RPA (±Ratios)

Figure 4 brings together AUROC analysis of different sPAP values (sPAP 40–45–50 mmHg) in combination with RPA and its ratios indexed an AA diameter or BSA or both.

Similar to Figure 3 with the main pulmonary artery, the highest diagnostic quality of AUROC results was observed with the isolated right pulmonary artery at an sPAP ≥ 40 mmHg (Figure 4A). RPA/BSA again stood out with the highest AUC (cut off value 15.16 mm/m^2^; AUC 0.676; 95% CI 0.569–0.783; *p* = 0.003; YI 0.39; sensitivity 0.63; specificity 0.76). However, no CT morphological parameter reached an AUC ≥ 0.700 in this case.

With an sPAP ≥ 45 mmHg (Figure 4B) and 50 mmHg (Figure 4C), respectively, the best results were again seen in the RPA/BSA ratio but remained behind those with an sPAP ≥ 40 mmHg.

### 3.4. AUROC Results—sPAP and LPA (±Ratios)

A direct comparison between the AUROC results of the right and left pulmonary artery can be achieved by comparing Figure 4 and Figure 5.

Here, it is noticeable that the AUROC results of LPA behave similarly to those of RPA. The ratio of LPA/BSA remains the leading combined CT parameter for predicting sPAP ≥ 40 mmHg (Figure 5A), ≥45 mmHg (Figure 5B), and ≥50 mmHg (Figure 5C).

The only difference from the previous results is that the highest AUC for the LPA/BSA ratio in this case is not found at an sPAP ≥ 40 mmHg but at an sPAP ≥ 50 mmHg (cut off value 15.03 mm/m^2^; AUC 0.698; 95% CI 0.569–0.827; *p* = 0.002; YI 0.45; sensitivity 0.63; specificity 0.82).

### 3.5. AUROC Results—sPAP and AA (±Ratios)

In addition, the diameter of the ascending aorta was also compared with different sPAP values (Figure 6). Here, the expected missing correlations became apparent with average AUC values ranging from 0.492 to 0.605. In each case, a slight improvement of the AUROC residual rates could be achieved by indexing the diameter of the ascending aorta to the BSA.

### 3.6. Correlation Analysis

Finally, to investigate relationships between different sPAP values (sPAP 40–45–50 mmHg) and CT morphological criteria of pulmonary hypertension, Spearman correlation analysis was performed (Table 3).

The best positive correlation was found between sPAP ≥ 40 mmHg and PA/BSA ratio with a Spearman correlation coefficient of 0.408 and a *p* < 0.001. This dropped significantly in the further course to sPAP ≥ 45 mmHg and ≥50 mmHg. A comparable correlation coefficient of 0.398 was exhibited by the ratio of PA/AA/BSA at an sPAP ≥ 40 mmHg. The correlations in which the isolated right or left pulmonary artery was involved never reached a value ≥ 0.300 with regard to the correlation coefficients.

## 4. Discussion

### 4.1. sPAP 40 mmHg with the Best Results—Pathophysiological Attempt of Explanations

At an sPAP of ≥40 mmHg, this study showed the best results regarding agreement between CT-derived vascular parameters and echocardiography in both, AUROC results and correlation analyses. Accordingly, a further increase of pressure in the pulmonary circulation with, for example, an sPAP ≥ 45 mmHg or ≥50 mmHg did not lead to a further increase in the vessel diameter of pulmonary trunk or right or left pulmonary artery, respectively (compare Table 2).

Reasons for this could potentially be explained by the following pathophysiological mechanisms: In the pulmonary trunk as well as the central pulmonary arteries, chronic pressure load leads to increased fragmentation of the elastic fibers [17]. This irreversible destruction of elastin components leads primarily to dilatation of the vascular structures and to the occurrence of inflammatory reactions. These initiated inflammatory processes in turn cause an invasion of fibroblasts, which leads to an imbalance between elastic and collagenous components due to a relevant collagen production and thus causes vascular stiffening [18,19]. A further increase of pressure in the small vessel does not lead to a further dilatation of the vessel wall, because the collagenous remodeling processes do not yield to the pressure, but rather the inflammatory reactions lead to a further collagen production and thus to a reduction of the inner diameter of the vessel. According to the physically simplified law of Hagen–Poiseuille, a decrease of the inner vessel radius leads to a significant increase of the vessel resistance (R = 1/r^4^).

### 4.2. PA vs. LPA and RPA—Why Do the Pulmonary Arteries Perform Worse Compared to the Pulmonary Trunk?

A direct comparison of the diagnostic significance and agreement of PA, LPA, and RPA with respect to different sPAP values provides a superiority of the PA diameter compared with both pulmonary arteries not only with respect to AUROC results but also with respect to correlation analyses. These results were almost congruent with a work of Rehman et al. [20], who compared the respective echocardiographic sPAP values with the PA, RPA, and LPA diameters. This showed very weak correlations between the sPAP and the right (r = 0.155) and left pulmonary artery (r = 0.138) diameters and a weak correlation between the sPAP and PA diameters (r = 0.316). Thus, simultaneously as in the present work, the LPA diameter was inferior to the RPA diameter in that of Rehman et al.

Reasons for this constellation can only be speculated, but are most likely to be attributed to anatomical conditions. Since the left lung, in contrast to the right lung, has only two lobes, the left pulmonary artery is in most cases the vessel with the smaller caliber. This could also be demonstrated in this work purely descriptively on the basis of the baseline characteristics, since the right pulmonary artery with a diameter of 27.55 mm has a larger caliber diameter on average by 2.1 mm than the left pulmonary artery with a diameter of 25.45 mm. Thus, already in the physiological state, there are different baseline positions between left and right pulmonary artery, which are additionally dependent on body size and weight. Accordingly, indexing to the BSA improved the diagnostic quality, but did not reach the significance of the pulmonary trunk by far.

The diameter of the AA is already pathophysiologically not consistent with PH, but rather with generalized arterial hypertension. Accordingly, the almost absent correlations to echocardiographic sPAP can be interpreted.

### 4.3. Body Surface Area—Significantly Improved Informative Value through Indexing

In the 2015 ESC guidelines [6], a pulmonary artery diameter ≥ 29 mm is seen as a potential indication for the presence of pulmonary hypertension regardless of genesis. The newly released August 2022 ESC guidelines [7] mention a combination of PA diameter ≥ 30 mm, a right ventricular outflow tract (RVOT) wall thickness ≥ 6 mm, and a septal deviation ≥ 140° as highly predictive regarding the presence of PH. Finally, numerous cutoff values regarding PA also based on different etiologies circulate in the literature. In a review work by Ussavarungsi et al. [21] in which PA diameters of the five different WHO groups of PH were investigated, cut-off values between 25.0 and 33.3 mm were shown for the detection of PH. In our study, the PA values settled at 29.50 mm, exactly between the proposed values of the 2015 and 2022 ESC guidelines.

In this work, however, it is clear that almost every measurement performed on CT associated with pulmonary hypertension leads to improved significance and thus diagnostic quality by indexing to the patient’s individually calculated body surface area.

To ultimately reconcile this discrepancy and strengthen the power of noninvasive radiological imaging, numerous papers are now moving toward indexing CT-derived pulmonary hypertension parameters to BSA [22,23]. Regarding patients with severe AS before TAVR, data are scarce. In this regard, Sudo et al. [24] were the only ones to demonstrate in a TAVR collective of 770 patients that the PA/BSA ratio among several CT-derived, vascular parameters provided the best AUROC results with an average of 16.80 mm/m^2^, with an AUC of 0.750. In the present study, the cut off ranged from 14.84 mm/m^2^ to 16.65 mm/m^2^, which was slightly lower. Simultaneously to the results of Sudo et al., the diagnostic predictive value of the pulmonary trunk was also significantly increased in our data after indexing to the BSA and also showed the best values in the correlation analyses (r = 0.408), which is why the BSA indexing should also be used for future, CT-morphological parameters with regard to an optimized statement.

## 5. Conclusions

Features related to pulmonary hypertension are fast and easily measurable on pre-TAVR CT and offer great potential regarding non-invasive screening for and detection of pulmonary hypertension in patients with severe AS. This knowledge can support and optimize the echocardiographic diagnosis since right heart catheterization is no longer a routine treatment. In this study, the diameter of the main pulmonary artery with the additionally determined ratios was superior to the values of the right and left pulmonary artery. Additional indexing to body surface area and thus further individualization of the parameters with respect to height and weight can further improve the diagnostic quality.

## 6. Limitation

The present study is based on data from a small cohort (n = 138) over a circumscribed time period (2016–2018) in two medical centers. Technical pitfalls in echocardiographic (for example due to patient-related limited ultrasound quality or due to difficult view on the tricuspid valve) and radiological measurements which lead to misclassifications should always be conceded, even if examinations were performed by experienced clinical investigators.

## Figures and Tables

**Figure 1 diagnostics-12-02363-f001:**
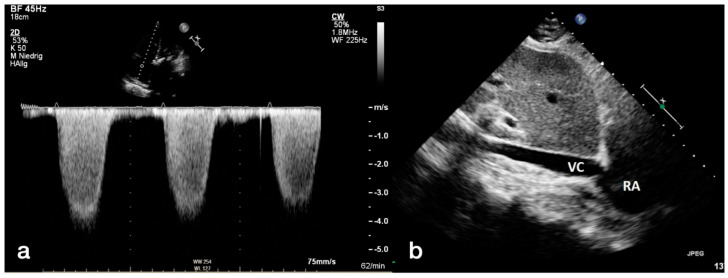
Measurement of systolic pulmonary artery pressure (**a**,**b**) with transthoracic echocardiography. Using continuous wave Doppler (**a**) over the tricuspid valve, the maximum regurgitation velocity (TRVmax) was determined, and pulmonary artery pressure was recorded. To obtain the systolic pulmonary artery pressure (sPAP), the diastolic diameter of the inferior vena cava was measured (**b**). RA: right atrium; VC: vena cava.

**Figure 2 diagnostics-12-02363-f002:**
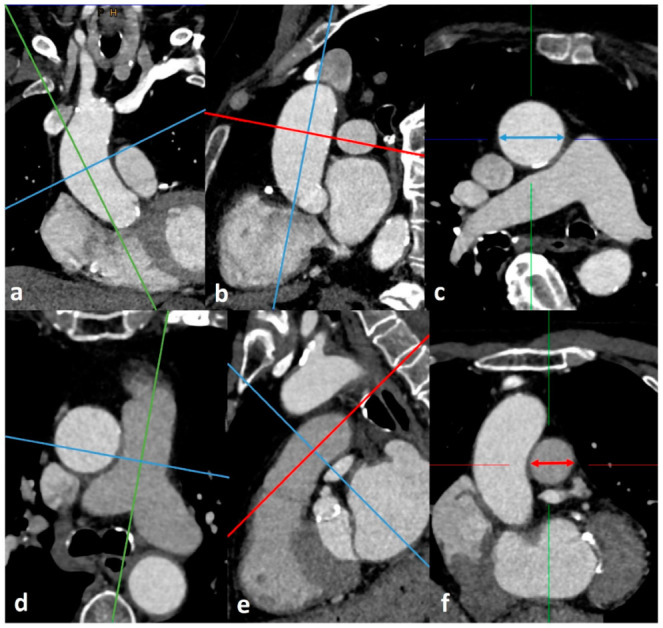
Measurement of ascending aorta (**a**–**c**) and main pulmonary artery (**d**–**f**) diameters at the level of the pulmonary artery bifurcation using the Extended Multiplanar Reconstruction Plugin in IMPAX. Planes were manually corrected perpendicular to the centerline of ascending aorta (**a**,**b**) and main pulmonary artery (**d**,**e**) using double-oblique MPR images. The thus generated axial vessel cross sections (**c**,**f**) were measured (blue double headed arrow in (**c**), red double headed arrow in (**f**), and the widest diameter was used as maximum diameter of AA and PA, respectively. RPA and LPA diameter were measured using the same method.

**Figure 3 diagnostics-12-02363-f003:**
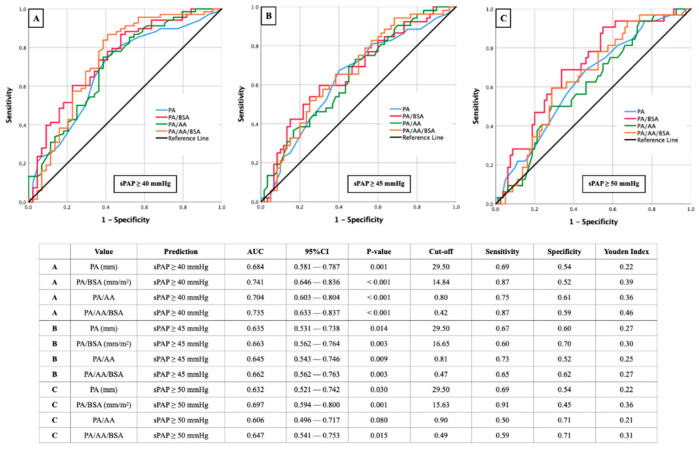
AUROC analyses with separately calculated cut-off values and Youden Index of PA (±ratios) for prediction of different sPAP values. (**A**): sPAP 40 mmHg; (**B**): sPAP 45 mmHg; (**C**): sPAP 50 mmHg. sPAP: systolic pulmonary artery pressure; PA: pulmonary artery; AA: ascending aorta; BSA: body surface area.

**Figure 4 diagnostics-12-02363-f004:**
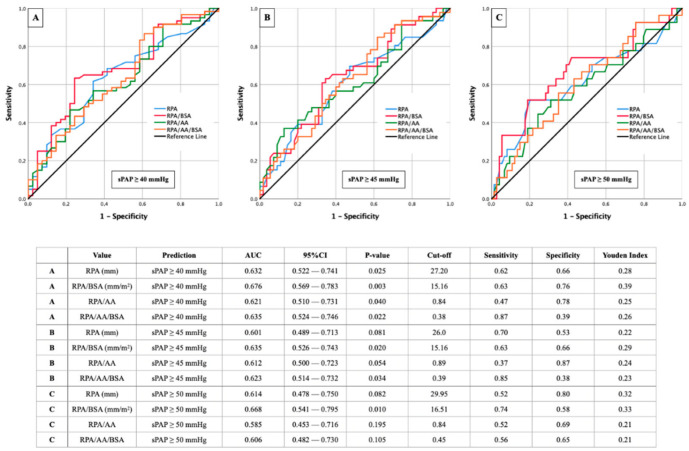
AUROC analyses with separately calculated cut-off values and Youden Index of RPA (±ratios) for prediction of different sPAP values. (**A**): sPAP 40 mmHg; (**B**): sPAP 45 mmHg; (**C**): sPAP 50 mmHg. sPAP: systolic pulmonary artery pressure; RPA: right pulmonary artery; AA: ascending aorta; BSA: body surface area.

**Figure 5 diagnostics-12-02363-f005:**
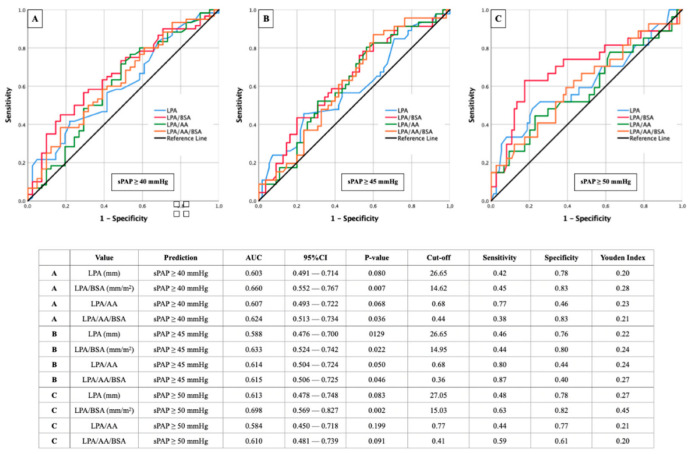
AUROC analyses with separately calculated cut-off values and Youden Index of LPA (±ratios) for prediction of different sPAP values. (**A**): sPAP 40 mmHg; (**B**): sPAP 45 mmHg; (**C**): sPAP 50 mmHg. sPAP: systolic pulmonary artery pressure; LPA: left pulmonary artery; AA: ascending aorta; BSA: body surface area.

**Figure 6 diagnostics-12-02363-f006:**
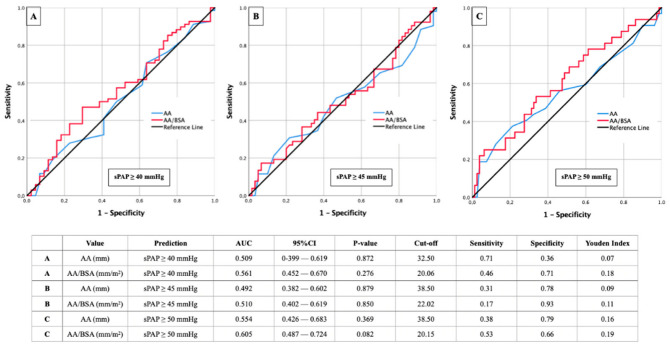
AUROC analyses with separately calculated cut-off values and Youden Index of AA (±ratios) for prediction of different sPAP values. (**A**): sPAP 40 mmHg; (**B**): sPAP 45 mmHg; (**C**): sPAP 50 mmHg.. sPAP: systolic pulmonary artery pressure; AA: ascending aorta; BSA: body surface area.

**Table 1 diagnostics-12-02363-t001:** Baseline characteristics of study population.

	Overall Cohort n = 138
Clinical Data	
Age (years)–mean ± SD	82.98 ± 5.05
Gender (male)–%	52.2
Weight (kg)–mean ± SD	73.55 ± 12.90
Height (cm)–mean ± SD	167.44 ± 8.67
BMI (kg/m^2^)–mean ± SD	26.21 ± 4.07
BSA (m^2^)–mean ± SD	1.82 ± 0.17
NYHA–median ± IQR	3.00 ± 0.50
STSScore–mean ± SD	3.01 ± 1.50
Concomitant Disease	
Diabetes mellitus–%	22.5
Arterial Hypertension–%	81.9
CVD–%	69.6
CVD–1 vessel–%	19.6
CVD–2 vessels–%	13.8
CVD–3 vessels–%	14.5
Myocardial infarction–%	3.6
Atrial fibrillation–%	34.1
Pacemaker–%	5.1
Malignancy–%	20.3
Stroke–%	7.2
PAD–%	6.5
COPD–%	8.0
Echocardiography	
LVEF (%)–mean ± SD	53.92 ± 10.32
LVEDD (mm)–mean ± SD	46.76 ± 6.18
AV Vmax (m/s)–mean ± SD	4.41 ± 0.52
AV dPmean (mmHg)–mean ± SD	49.76 ± 11.80
AV dPmax (mmHg)–mean ± SD	80.71 ± 18.80
sPAP (mmHg)–mean ± SD	44.94 ± 16.65
sPAP ≥ 40 mmHg–%	63.0
sPAP ≥ 45 mmHg–%	50.0
sPAP ≥ 50 mmHg–%	33.3
TAPSE (mm)–mean ± SD	22.01 ± 3.72
AVI ≥ II°–%	16.7
MVI ≥ II°–%	31.2
TVI ≥ II°–%	21.7
Computed Tomography	
PA ≥ 29 mm–%	60.9
PA/AA ≥ 0.80–%	62.9
PA/AA ≥ 0.90–%	35.7
Laboratory Data	
Crea (mg/dL)–median ± IQR	1.00 ± 0.40
BNP (pg/mL)–median ± IQR	1797.00 ± 2978.60
cTnI (pg/mL)–median ± IQR	22.00 ± 30.00
Hkt (%)–median ± IQR	39.00 ± 7.35
Hb (g/dL)–median ± IQR	12.90 ± 2.30
CK (U/L)–median ± IQR	74.00 ± 82.50

BMI: body mass index; BSA: body surface area; CVD: cardiovascular disease; PAD: peripheral artery disease; COPD: chronic obstructive pulmonary disease; LVEF: left ventricular ejection fraction; LVEDD: left ventricular end-diastolic diameter; AV Vmax: maximal velocity over aortic valve; AV dpmean: mean pressure gradient over aortic valve; AV dpmax: maximal pressure gradient over aortic valve; sPAP: systolic pulmonary artery pressure; TAPSE: tricuspid annular plane systolic excursion; AVI: aortic valve insufficiency; MVI: mitral valve insufficiency; TVI: tricuspid valve insufficiency; PA: pulmonary artery; AA: ascending aorta; Crea: creatinine; BNP: brain natriuretic peptide; cTnI: cardiac Troponin I; Hkt: hematocrit; Hb: hemoglobin; CK: creatine kinase; SD: standard deviation; IQR: interquartile range.

**Table 2 diagnostics-12-02363-t002:** CT morphological vascular parameters measured via computed tomography in dependence of different sPAP values.

	Overall Cohort n = 138	sPAP ≥ 40 mmHg n = 87	sPAP ≥ 45 mmHg n = 69	sPAP ≥ 50 mmHg n = 46
**Clinical Data**				
PA (mm)–mean ± SD	29.89 ± 5.45	31.19 ± 5.41	30.98 ± 5.22	31.56 ± 5.35
AA (mm)–mean ± SD	35.10 ± 4.61	35.12 ± 4.58	35.04 ± 4.94	35.84 ± 5.39
PA/BSA (mm/m^2^)–mean ± SD	16.51 ± 3.00	17.33 ± 2.52	17.22 ± 2.53	17.72 ± 2.36
PA/AA–mean ± SD	0.86 ± 0.13	0.89 ± 0.13	0.89 ± 0.13	0.89 ± 0.12
PA/AA/BSA–mean ± SD	0.47 ± 0.08	0.50 ± 0.07	0.50 ± 0.08	0.50 ± 0.07
RPA (mm)–mean ± SD	27.55 ± 4.59	28.40 ± 4.80	28.53 ± 5.06	28.98 ± 5.33
RPA/BSA (mm/m^2^)–mean ± SD	15.15 ± 2.45	15.68 ± 2.41	15.71 ± 2.46	16.06+ 2.56
RPA/AA–mean ± SD	0.79 ± 0.13	0.81 ± 0.13	0.82 ± 0.14	0.81 ± 0.13
RPA/AA/BSA–mean ± SD	0.44 ± 0.08	0.45 ± 0.08	0.46 ± 0.08	0.46 ± 0.08
LPA (mm)–mean ± SD	25.45 ± 3.43	26.00 ± 3.58	26.12 ± 3.73	26.66 ± 3.97
LPA/BSA (mm/m^2^)–mean ± SD	14.01 ± 1.93	14.39 ± 2.00	14.42 ± 2.00	14.84 ± 2.27
LPA/AA–mean ± SD	0.73 ± 0.10	0.75 ± 0.10	0.75 ± 0.10	0.75 ± 0.12
LPA/AA/BSA–mean ± SD	0.40 ± 0.07	0.42 ± 0.07	0.42 ± 0.07	0.43 ± 0.08
AA/BSA (mm/m^2^)–mean ± SD	19.43 ± 2.64	19.59 ± 2.48	19.52 ± 2.68	20.19 ± 2.85

sPAP: systolic pulmonary artery pressure; PA: pulmonary artery; AA: ascending aorta; RPA: right pulmonary artery; LPA: left pulmonary artery; BSA: body surface area.

**Table 3 diagnostics-12-02363-t003:** Tabular overview of Spearman correlation analysis with regard to different sPAP values (sPAP 40–45–50 mmHg) and CT morphological criteria of pulmonary hypertension.

Spearman Correlation	sPAP ≥ 40 mmHg		sPAP ≥ 45 mmHg		sPAP ≥ 50 mmHg	
	rs	*p*	rs	*p*	rs	*p*
PA	0.305	0.001	0.229	0.014	0.203	0.029
PA/BSA	0.408	<0.001	0.282	0.003	0.309	0.001
PA/AA	0.324	<0.001	0.236	0.011	0.156	0.096
PA/AA/BSA	0.398	<0.001	0.281	0.003	0.230	0.014
RPA	0.239	0.015	0.191	0.052	0.193	0.050
RPA/BSA	0.291	0.003	0.220	0.025	0.239	0.015
RPA/AA	0.208	0.036	0.194	0.051	0.132	0.187
RPA/AA/BSA	0.230	0.021	0.212	0.033	0.162	0.106
LPA	0.184	0.062	0.160	0.106	0.178	0.070
LPA/BSA	0.246	0.012	0.200	0.043	0.252	0.010
LPA/AA	0.189	0.057	0.202	0.041	0.132	0.186
LPA/AA/BSA	0.210	0.035	0.199	0.046	0.169	0.091
AA	0.023	0.807	−0.008	0.933	0.087	0.353
AA/BSA	0.103	0.278	0.018	0.851	0.165	0.082

sPAP: systolic pulmonary artery pressure; PA: pulmonary artery; AA: ascending aorta; RPA: right pulmonary artery; LPA: left pulmonary artery; BSA: body surface area; rs: correlation coefficient of Spearman.

## Data Availability

The data presented in this study are available on request from the corresponding author.

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
