# Peer review of "Comparability between Computed Tomography Morphological Vascular Parameters and Echocardiography for the Assessment of Pulmonary Hypertension in Patients with Severe Aortic Valve Stenosis—Results of a Multi-Center Study"

_diagnostics, 2022, doi:10.3390/diagnostics12102363_

Round 1

Reviewer 1 Report

Authors of the article compared morphological vascular parameters available in CT scans and echocardiography for the assessment of pulmonary hypertension in patients with severe aortic valve stenosis. It was a pleasure to review this article because of its undeniable strengths:
- professional English language used in the manuscript regarding the discussed issue; 
- clearly emphasised inclusion & exclusion criteria for the study group;
- study was conducted in accordance to principles of the Declaration of Helsinki and Good Clinical Practice with informed consent obtained from all of the members of the study group;

- very advanced statistical analysis with legible presentation and comprehensive discussion.

Only one suggestion appeared with regard to the manuscript: authors have prepared 22 publications in the reference section. Some of them are dated back to 2008. It would be better to moderate and expand the reference list with publications dating up to 5-6 years ago. It would improve the literature knowledge concerning the work.

Sincere congratulations on your research!

Author Response

Dear Reviewer 1,

thank you very much for taking your time and for reading our manuscript.

The comments contribute a lot to our continuous improvement.

You will find corresponding answers to your suggestions and questions point-by-point in a separate PDF file.

Thank you very much for the recent proofreading!

Kind regards,

Elke Boxhammer

Reviewer 2 Report

1) Introduction. L 58-59.In addition to accurate preoperative/preinterventional planning, CT diagnostics can  also provide evidence for the presence of pulmonary hypertension (PH) in patients with  severe AS 3. Please improve this sentence and add these references:

a- Diagnosis and Treatment of Pulmonary Arterial Hypertension: A Review. JAMA. 2022 Apr 12;327(14):1379-1391. doi: 10.1001/jama.2022.4402. 

b- Chronic Thromboembolic Pulmonary Hypertension: An Observational Study. Medicina (Kaunas). 2022 Aug 13;58(8):1094. doi: 10.3390/medicina58081094. 

2) Introduction. L75-78. Thus, the aim of the retrospective multicenter study design presented here was to compare CT morphological vascular parameters potentially associated with PH with different echocardiographically obtained sPAP values in order to draw further noninvasive  conclusions about PH in TAVR patients. Please improve the description of study aim.

3) Methods. Transthoracic echocardiography. Please add a figure of Transthoracic echocardiography.

4) Figure 1. L 137-142. Measurement of ascending aorta (a-c) and main pulmonary artery (d-f) diameters at the level of the pulmonary artery bifurcation using the Extended Multiplanar Reconstruction Plugin in IMPAX. Planes were manually corrected perpendicular to the centerline of ascending aorta (a,b) and main pulmonary artery (d,e) using double-oblique MPR images. The thus generated axial vessel cross sections (c, f) were measured in long  and short axis (blue and green double headed arrow in (c), red and green double headed arrow in (f)), and the widest diameter was used as maximum diameter of AA and MPA respectivley. RPA and LPA diameter were  measured using the same method. Please, improve the quality of this figure.

5) Results 161 Baseline Characteristics. Please, underline in the text the most important statistical values to support the results.

6) Discussion L256-262. sPAP 40 mmHg with the best results — pathophysiological attempt of explanations  At an sPAP of ≥ 40 mmHg, this study showed the best results regarding agreement  between CT-derived vascular parameters and echocardiography in both, AUROC results and correlation analyses. Accordingly, a further increase of pressure in the pulmonary  circulation with, for example, an sPAP ≥ 45 mmHg or ≥ 50 mmHg did not lead to a further increase in the vessel diameter of pulmonary trunk or right or left pumonary artery, re-spectively. Please, clarify the most important statistical values.

7) Conclusion L327-333. Features related to pulmonary hypertension are fast and easily measureable on pre-  TAVR CT and offer great potential regarding non-invasive screening for and detection of  pulmonary hypertension in patients with severe AS and can support the echocardio-graphic diagnosis. In this study, the diameter of the mean pulmonary artery with the ad-ditionally determined ratios were superior to the values of the right and left pulmonary artery. Additional indexing to body surface area and thus further individualization of the parameters with respect to height and weight can further improve the diagnostic quality. Please, improve the conclusions and underline the novelty and the clinical implications of the study.

8) Please, underline the limits of the study and improve this section.

Author Response

Dear Reviewer 2,

thank you very much for taking your time and for reading our manuscript.

The comments contribute a lot to our continuous improvement.

You will find corresponding answers to your suggestions and questions point-by-point in a separate PDF file.

Thank you very much for the recent proofreading!

Kind regards,

Elke Boxhammer
